# δ-Catenin Participates in EGF/AKT/p21^Waf^ Signaling and Induces Prostate Cancer Cell Proliferation and Invasion

**DOI:** 10.3390/ijms22105306

**Published:** 2021-05-18

**Authors:** Yingjie Shen, Hyoung Jae Lee, Rui Zhou, Hangun Kim, Gen Chen, Young-Chang Cho, Kwonseop Kim

**Affiliations:** 1College of Pharmacy and Research Institute of Pharmaceutical Sciences, Chonnam National University, Gwangju 61186, Korea; syj583902539@163.com (Y.S.); ableleehj@gmail.com (H.J.L.); chengen86374062@126.com (G.C.); yccho@jnu.ac.kr (Y.-C.C.); 2College of Pharmacy and Research Institute of Life and Pharmaceutical Sciences, Sunchon National University, Sunchon 57922, Korea; zhourui274@gmail.com (R.Z.); hangunkim@scnu.ac.kr (H.K.)

**Keywords:** δ-catenin, prostate cancer, EGF, AKT, p21^Waf^

## Abstract

Prostate cancer (PCa) is the second most leading cause of death in males. Our previous studies have demonstrated that δ-catenin plays an important role in prostate cancer progression. However, the molecular mechanism underlying the regulation of δ-catenin has not been fully explored yet. In the present study, we found that δ-catenin could induce phosphorylation of p21^Waf^ and stabilize p21 in the cytoplasm, thus blocking its nuclear accumulation for the first time. We also found that δ-catenin could regulate the interaction between AKT and p21, leading to phosphorylation of p21 at Thr-145 residue. Finally, EGF was found to be a key factor upstream of AKT/δ-catenin/p21 for promoting proliferation and metastasis in prostate cancer. Our findings provide new insights into molecular controls of EGF and the development of potential therapeutics targeting δ-catenin to control prostate cancer progression.

## 1. Introduction

The prostate is a small muscular gland locates in front of the rectum and under the bladder in the man’s pelvic cavity behind the pubis [1]. According to relevant statistical analysis results, prostate cancer (PCa) is the second most leading cause of cancer-associated deaths in men [2]. The number of new prostate cancer cases in 2020 was estimated to be 191,930 globally [3]. Genetics, family history, chronic inflammation, advanced age, diet, and environment are well-established risk factors for PCa development [4,5]. However, relevant pathways accounting for PCa development are not fully clarified yet. δ-Catenin was found that the gene designation for catenin of the delta subfamily already existed for p120ctn [6]. Therefore, this subfamily now has two members: CTNND1 for p120ctn and CTNND2 for δ-catenin/NPRAP [7]. However, little is known about δ-catenin’s signaling network. In most cases, δ-catenin can bind to the juxtamembrane domain region of E-cadherin, a calcium-regulated cell–cell junction protein [7,8]. However, more and more studies have pointed out that δ-catenin also plays other critical roles in cell physiology besides its role in cytoskeleton remodeling. For example, δ-catenin expression begins at early embryogenesis [2]. Meanwhile, when δ-catenin is abundantly expressed, it can cause down-regulation of dendrogenesis and cognitive functions [9]. The relationship between δ-catenin and cancer has also been studied extensively, especially in prostate cancer. δ-Catenin is overexpressed in roughly 11% of prostate cancers [10]. It has been described as a potential diagnostic biomarker [11,12]. Moreover, it has been reported that δ-catenin mRNA is overexpressed in prostate cancer than in benign prostate hyperplasia (BPH) [6]. The overexpression of δ-catenin can promote PC cell growth and progression by inducing the cell proliferation and profiles of survival genes [13]. However, its molecular mechanism remains unclear. Understanding how δ-catenin regulates prostate cancer can provide some strategies for the clinical treatment of the prostate.

P21, also known as wild-type p53-activated fragment 1 (WAF1), is one of the cell cycle regulatory proteins [14,15]. It plays key roles in various cellular events, such as apoptosis, cell migration, DNA repair, tumor formation [16]. In response to DNA damage and other cellular stressors, p21 expression can be increased, resulting in the activation of cell cycle checkpoints until repair takes place [17]. Since its expression change has been reported in a broad range of cancers, it is considered an indicator of cancer [16]. However, recent research studies have revealed that p21 not only functions as a tumor suppressor but also acts as an oncogene with a dual behavior in different cellular locations [18,19]. In many cancer types, p21 is stabilized in the cytoplasm to inhibit its nucleus translocation. Accumulation of p21 in the cytoplasm can induce cell proliferation and suppresses apoptosis, resulting in resistance to chemotherapy and radiotherapy [14]. Our study reported here revealed that δ-catenin could stabilize p21 in the cytoplasm of prostate cancer cells, thus blocking the nuclear accumulation of p21.

Epidermal growth factor (EGF) plays an important role in controlling the physiological function of the prostate. When EGF binds to EGF receptor (EGFR), it can induce cell growth, proliferation, and migration of different tumors, including PCa [20]. In metastatic PCa, EGF/EGFR usually shows aberrant expression, which is closely related to poor prognosis and low survival rate [21]. Regarding the molecular mechanism, EGF in PCa cells can induce epithelial-mesenchymal transition through protein kinase C (PKC)/GSK3β/Snail signaling pathway [22]. Meanwhile, EGF can also promote the stability of upregulated Snail by inducing PKC activation, consequently preventing GSK-3β phosphorylation activity, leading to decreased Snail ubiquitination and increased the Snail’s transcription [23]. In prostate cancer, δ-catenin can stabilize EGFR to enhance EGF signaling. In addition, EGF can increase δ-catenin phosphorylation [24]. However, some underlying signaling mechanisms for the interaction between EGF and δ-catenin remain unclear.

Considering the function of EGF in prostate cancer, uncovering the mechanism of δ-catenin involved in EGF regulation is particularly important. In the present study, we investigated whether δ-catenin could block p21 nuclear accumulation by stabilizing cytoplasm p21. Results of this study showed that cytoplasmic p21 was regulated by its Thr145 phosphorylation. We also found that δ-catenin could induce the interaction between AKT and p21, which increased the phosphorylation of p21. Moreover, we found that EGF was one of the upstream ligands for regulating AKT/δ-catenin signaling in prostate cancer.

## 2. Results

### 2.1. δ-Catenin Increases p21 Protein Level by Increasing Its Stabilization

Our previous data have shown that δ-catenin can induce the proliferation, migration, and invasion of prostate cancer [25]. However, the underlying mechanism involved is unclear. It is well-known that p21 is one of the tumor suppressors, and it is usually be used as a biomarker to indicate tumor procession [26]. At the beginning of this study, the function of δ-catenin in p21 regulation was investigated to confirm the effect of δ-catenin in promoting prostate tumor. We detected p21 protein levels after overexpressing δ-catenin. After we transfected GFP-δ-catenin full length (FL) and GFP into CWR22Rv-1 (Rv-1) cell line, Western blot (WB) was performed to determine the transfection efficiency (Appendix A). Contrary to the expected results, p21 protein was induced by δ-catenin (Figure 1a,b). This suggests that there might be other ways for δ-catenin to regulate p21. To understand how δ-catenin could affect p21, we detected mRNA levels of p21 using Real-Time Quantitative Reverse Transcription (qRT-PCR). The mRNA level of p21 was mildly decreased when we overexpressed δ-catenin. Clearly, the mRNA level of p21 was not consistent with its protein level (Appendix A). p21 is an unstable protein with a relatively short half-life [27]. Thus, we hypothesized that δ-catenin could induce p21 stabilization. To test this hypothesis, we used a protein synthesis inhibitor (CHX) to treat Rv-1 cells and harvested cells at different time points (Figure 1c). We then used a single exponential decay model for data analysis with the following equation: N(t) = N(0)e^−λt^ (Figure 1d). The exponential decay constant (λ) was much smaller when Rv-1 cells were co-treated with δ-catenin and CHX. The half-life of p21 was also predicted with the model. The group with δ-catenin overexpression showed a longer half-life, suggesting that δ-catenin could stabilize p21 (Figure 1e). Our results provided compelling evidence that δ-catenin can stabilize p21, leading to increased p21 protein levels in Rv-1 cells.

### 2.2. δ-Catenin Blocks p21 Nuclear Accumulation by Stabilizing p21 in the Cytoplasm

The CDK inhibitor p21, also known as p21^waf1/cip1^, is a well-known inhibitor of the cell cycle. It can arrest cell cycle progression in G1/S and G2/M transitions by inhibiting CDK4, 6/cyclin-D and CDK2/cyclin-E, respectively [28]. Despite its ability to inhibit the cell cycle, recent studies have reported that cytoplasmic p21 can bind to proteins involved in the induction of apoptosis, thus inhibiting the activities of such proteins [29]. As a tumor inducer, δ-catenin has been found to show high levels in different cancers, including PCa [10]. Combining with the phenomenon that δ-catenin can stabilize p21, it seems plausible that δ-catenin can retain p21 in the cytoplasm. To test this possibility, at 24 h after transfection, we separated nuclear proteins and found that p21 protein levels were reduced in nuclear extracts of cells overexpressing δ-catenin than in normal cells (Figure 2a,b). Immunofluorescence staining was also performed to analyze p21 nuclear accumulation in δ-catenin overexpressed cells. Results showed that δ-catenin could hinder p21 from getting into the nucleus (Figure 2c). Several studies have suggested that phosphorylation of p21 at Thr-145 residue renders p21 stable in the cytoplasm [14,19]. Thr-145 is located in the vicinity of the nuclear localization sequence (NLS), the phosphorylation of which can prevent interactions of p21 with importins and inhibits nucleus translocation of p21. To further investigate the effect of δ-catenin on p21, we detected phosphorylation of p21 at Thr-145 residue in δ-catenin-overexpressed Rv-1 cells. Results of Western blot revealed that δ-catenin could induce the phosphorylation of p21 on Thr-145 (Figure 2d,e). Following plasmid transfection, si-RNA of δ-catenin was used for treatment to exclude the relationship between δ-catenin and p21. Results revealed that silencing δ-catenin in Rv-1 cells by siRNA also showed a strong promoted effect on nuclear accumulation of p21(Appendix A) and blocked the phosphorylation of p21 (Appendix A). Taken together, these findings suggest that δ-catenin can block p21 nuclear translocation by promoting phosphorylation of p21 at Thr-145 residue.

### 2.3. δ-Catenin Can Promote the Interaction between p21 and AKT Which Phosphorylates p21 at Thr-145

In light of the prominent phosphorylation of p21 in prostate cancer, we next focused on the upstream regulator of p21. Some studies have shown that cytoplasmic p21 phosphorylated by AKT pathway at Thr-145 residue can inhibit caspase3 and CDK2. LY294002(LY) is regarded as an inhibitor of PI3K. Since PI3K is an upstream kinase of AKT, LY is also used for inhibiting AKT phosphorylation in cells [30]. After Rv-1 cells were treated with LY, the phosphorylation of p21 at Thr-145 residue was significantly blocked. This demonstrated the pro-phosphorylation effect of AKT on p21 in prostate cancer (Appendix A). However, whether δ-catenin plays a role in AKT regulation of p21 remains unclear. Previously, we have demonstrated that AKT1 plays an important role in human diseases by phosphorylating δ-catenin at Thr-454 residue [10]. To further understand the functional abilities of δ-catenin in the interaction between AKT and p21, we used a site-directed δ-catenin mutant and a meaningless δ-catenin mutant T454A to transfect Rv-1 cells. WB was then performed to determine the transfection efficiency of T454A (Appendix A). Results of quantification revealed that phosphorylation of p21 was significantly decreased after transfection with T454A mutant (Figure 3a,b). We also performed immunoprecipitation to analyze the interaction between AKT and p21 in the context of δ-catenin. Results also showed that δ-catenin induced the interaction between AKT and p21. Conversely, after transfection with the T454A mutant, such interaction was blocked (Figure 3c). To further determine the regulation of δ-catenin, we used different deletion mutants of 1–690 and 691–1070. Immunoprecipitation results, a similar conclusion was made. The 691–1070 fragment mutant decreased the association between AKT and p21 (Figure 3d). We also used different deletion mutants of δ-catenin (1–690 and 691–1040) to transfect cells to analyze p21 cellular location. WB was used to determine the plasmid transfection efficiency (Appendix A). We found that δ-catenin could stabilize cytoplasm p21. However, such stabilization was blocked in the group transfected with the deletion mutant the lost the 1–690 fragment (Figure 3e,f). In addition, cells overexpressing δ-catenin exhibited the lowest level of p21 in the nucleus, while transfection of the T454A mutant reversed this effect (Appendix A). These results strongly suggest that δ-catenin can be induced by AKT, leading to p21 stabilization in the cytoplasm through AKT/p21 signaling.

### 2.4. EGF Is One of Upstream Ligands That Can Regulate AKT/δ-Catenin/p21 Signaling

There are many upstream proteins that can functionally drive the AKT signaling pathway. EGF is one of the important upstream signals in prostate cancer [31]. Previous studies have confirmed that EGFR and δ-catenin have a crosstalk with each other and that EGF can increase δ-catenin protein levels [24]. These previous results indicate that there might be some associations among EGF/EGFR, AKT, and δ-catenin. To uncover this fuzzy mechanism, gefitinib was used to block the function of EGFR in EGF-induced Rv-1 cells. Consisting with our previous results, gefitinib decreased the phosphorylation of AKT. Furthermore, gefitinib caused a strong inhibition of p21 phosphorylation, which deepened our understanding of the connections among EGF/EGFR, AKT and p21 (Appendix A). Moreover, Rv-1 cells were transfected with mutant δ-catenin. Transfected cells were then serum-starved for 24 h and treated with 100 ng/mL EGF for 30 min. Results showed that co-treatment with full-length δ-catenin and EGF resulted in the highest phosphorylation level of p21. On the contrary, the 691–1070 fragment mutant of δ-catenin and T454A mutant of δ-catenin blocked the increase of phosphorylation level of p21 caused by EGF. These results demonstrate that EGF can phosphorylate p21 through the AKT pathway and that δ-catenin takes part in EGF/AKT/p21 signaling (Figure 4a,b). We also found that losing δ-catenin downregulated EGF-induced phosphorylation of p21 (Appendix A). Immunoprecipitation was also performed to determine the relationship between AKT and p21 induced by EGF. As expected, co-treatment with EGF and δ-catenin further induced the interaction of AKT with p21 (Figure 4c). These results provide evidence that δ-catenin plays an important role in the regulation of p21 phosphorylation by EGF.

### 2.5. EGF Regulates Proliferation and Invasion of Cells through AKT/δ-Catenin/p21 Signaling Independently

Many studies have confirmed that EGF/EGFR signaling is one important growth factor signaling involved in the regulation, proliferation, and survival of prostate cancer [32,33]. However, little is known about how EGF borrows δ-catenin to enhance AKT/p21 signaling, which induces PCa proliferation and invasion. Accordingly, we first used CCK8 to determine cell proliferation. Cells were co-treated with mutant δ-catenin and EGF and then incubated with CCK8 solution for 6 h. Next, the absorbance of incubated cells was measured at 490 nm. Results showed that cells transfected with δ-catenin had much higher absorbance. Moreover, treatment with EGF resulted in a higher level of cell proliferation than the control group not treated with EGF. However, absorbance was decreased for cells transfected with the 691–1040 deletion mutant or the T454A mutant. Our results provide evidence that AKT/δ-catenin signaling is involved in the effect of EGF on cell proliferation and that the 1–690 fragment and the T454 site of δ-catenin play an important role in cell proliferation (Figure 5a). We also performed a wound-healing assay to determine the migration ability. Results showed that overexpression of δ-catenin or EGF induced PCa cells migration, whereas the 691–1040 mutant of δ-catenin and the T454A mutant of δ-catenin affected the pro-migration of EGF (Appendix A). As for invasion, a transwell assay was performed. Transfected cells were seeded onto the transwell membrane in the upper chamber, and serum-free medium or EGF-containing medium was added to the lower chamber. Results clearly showed that all groups treated with EGF had more cells. However, cells transfected with the 691–1040 mutant orT454A mutant of δ-catenin inhibited the function of EGF, leading to lower amounts of cells than groups treated with EGF (Figure 5b,c). Together, our findings provide compelling evidence that the pro-proliferation and pro-migration effect of EGF depends on the presence of 1–690 or T454 phosphorylation of δ-catenin that was closely regulated by AKT.

PC3 is another cell line that shows high levels of δ-catenin [9]. As expected, overexpressing δ-catenin blocked p21 nuclear accumulation, while cells transfected with T454A mutant showed the highest level of p21 in the nucleus (Appendix A). Moreover, interference with endogenous δ-catenin expression using si-RNA also affected the phosphorylation level of p21, resulting in the downregulation of its phosphorylation at Thr-145 residue (Appendix A). In the meantime, the CCK8 assay was used to determine the proliferation of PC3 cells overexpressing different δ-catenin mutants in a media containing EGF (Appendix A). Similarly, transwell assay results showed that both EGF and δ-catenin significantly promoted the invasive capacity of PC3 cells, whereas δ-catenin 691–1040 mutant and T454A mutant of δ-catenin blocked the pro-invasion effect of EGF (Appendix A). These results showed us the important role of δ-catenin phosphorylation at Thr-145 residue in the pro-proliferation effect of EGF.

Overall, these results are expected considering that previous results have shown that AKT/δ-catenin/p21 signaling plays an important role in the effect of EGF on cell proliferation and invasion.

## 3. Discussion

Prostate cancer is the second most common cause of cancer-related deaths in males [3]. Surgery is one of the valid tools for blocking the progression of localized prostate cancer. However, its cure rate and prognosis are poor. Thus, it is particularly important to find a new treatment method from the perspective of pathogenesis. Previous studies have reported that almost all prostate cancer-related progression cases are associated with the proliferation of cancer cells and distant metastasis [4]. Clinically, the generally recommended treatment for prostate cancer is androgen deprivation therapy (ADT). However, it has limited clinical outcomes and a high risk of recurrence [4]. Based on this fact, thorough knowledge about the molecular apparatus involved in prostate cancer proliferation and distant metastasis formation and knowledge about the signaling that regulates cancer progression is of critical importance to develop novel therapeutic approaches.

δ-catenin was first identified as a binding partner of presenilin-1 in 1997 [6]. It belongs to the p120-catenin(p120ctn) subfamily of armadillo protein [3]. Recently, more and more studies have found that δ-catenin plays an important regulatory role in different cancer and cognitive diseases [11]. Our previous study has also found the role of δ-catenin in regulating prostate cancer progression through E2F and Wnt pathways [9,34]. However, underlying signaling pathways of δ-catenin need further studies.

In the present study, when δ-catenin was overexpressed in Rv-1 cells, the p21 protein level was induced. A further study is needed to find the regulation of δ-catenin for p21 and the major target for the stabilization of p21. As a key transcription factor, p21 has dual roles: inhibiting cell cycles and inhibiting apoptosis. Such dual roles of p21 are due to its cellular location. When p21 enters into the nucleus, it will bind with key proteins of cell cycle checkpoints (CDK1/2) and block the cell cycle [35]. However, if p21 stays in the cytoplasm, it will bind with anti-apoptosis proteins and induce the stabilization of p21 [18]. To discover the mechanism involved in the effect of δ-catenin on p21, we determined the location of p21 and found that δ-catenin could block p21 from translocating to the nucleus. Subsequent research, we also confirmed that phosphorylation of p21 at Thr-145 residue affected by δ-catenin also affected the localization of p21 in the cytoplasm. Phosphorylation of Thr-145 is regulated by AKT signaling in a previous study [14]. Thus, δ-catenin might regulate AKT, thus affecting p21. To uncover details of δ-catenin, we transfected cells with δ-catenin mutants, one with a mutation at AKT target site Thr-454, another with deletion mutant of δ-catenin. As expected, when AKT target site mutant δ-catenin (T454A) was overexpressed, the interaction between AKT and p21 was decreased. Meanwhile, when we transfected cells with δ-catenin deletion mutant that deleted fragment of 691–1070, decreased interaction between AKT and p21 was observed, further confirming that fragment 1–690 of δ-catenin would play an important role in the regulation of AKT for p21. These results indicate that AKT can phosphorylate δ-catenin at the T454 site, which in turn activates AKT/p21 signaling and retains p21 in the cytoplasm.

AKT has many upstream signals, among which EGF/EGFR is one signal that is closely associated with tumor progression. We also found that δ-catenin could stabilize EGFR protein to enhance EGF signaling. As a member of growth factors, EGF is highly expressed in various cancers, including prostate cancer [21]. EGF causes rapid phosphorylation of EGFR to activate different intracellular pathways. Whether δ-catenin has other targets besides EGFR that participate in the regulation of EGF for tumors needs to be further explored. In our research, we discovered that EGF/EGFR could phosphorylate AKT and induce the phosphorylation of p21 at Thr-145 residue. However, increased phosphorylation of p21 by EGF/EGFR was blocked by overexpressing AKT target site mutant of δ-catenin (T454A) and a deletion mutant of δ-catenin (691–1040 fragment). Moreover, we found that δ-catenin could enhance the interaction between AKT and p21 in the presence of EGF. It is well-known that EGF is involved in cancer progression by promoting cell proliferation and invasion [36]. The present study provided evidence that pro-proliferation and pro-invasion effects of EGF depended on the whole length of δ-catenin as such effects of EGF were reversed by T454A mutant of δ-catenin and 691–1070 fragment deletion mutant of δ-catenin.

These data provide evidence for the role of EGF in cancer proliferation and invasion through AKT/δ-catenin/p21 signaling (Figure 6). We show an aberrant δ-catenin expression in PCa cells constitutive induces EGF/AKT signaling in pro-proliferation and pro-invasion. We also found that p21 was a key downstream factor stabilized in the cytoplasm to block its transcription. It will be intriguing to investigate whether we can target δ-catenin to block EGF signaling in prostate cancer patients. Almost all PC express the androgen receptor (AR) or AR variants, such as ARV7. Since EGFR can signal through AR to regulating cell proliferation and motility of PC cells [37], the influence of AR functions in EGF/EGFR signaling cannot be undervalued in PC. These observations can serve as the foundation for the future development of new therapeutics for prostate cancer. In follow-up clinical trials, we can investigate several approaches to target AKT/δ-catenin/p21 to attenuate EGF signaling in prostate cancer.

## 4. Materials and Methods

### 4.1. Inhibitors

Inhibitors were purchased from commercial companies as follows: PI3K inhibitor: LY294002 (Selleck, Houston, TX, USA); eukaryote protein synthesis inhibitor: CHX (Selleck, Houston, TX, USA); EGFR inhibitor: gefitinib (Selleck, Houston, TX, USA).

### 4.2. Plasmids

Constructs of δ-catenin wild-type (WT) in pEGFP-C1 and deletion constructs of 1–690, 690–1040, and T454A were generated by PCR amplification and cloned into a pEGFP-C1 vector as showed in previous research [34]. δ-catenin siRNA (SC43021) was purchased from Santa Cruz Biotechnology (Dallas, TX, USA).

### 4.3. Antibodies

The following antibodies were purchased from commercial companies: anti-δ-catenin (#611537, BD Bioscience, Franklin Lakes, NJ, USA), anti-p21^Waf1/Cip1^ (#SC6246, Santa Cruz Biotechnology, Dallas, TX, USA), anti-phospho-p21^Thr 145^ (#SC-377569, Santa Cruz Biotechnology, Dallas, TX, USA), anti-AKT (#4691, Cell Signaling Technology, Danvers, MA, USA), anti-phospho-AKT^Ser 473^ (#4060, Cell Signaling Technology, Danvers, MA, USA), anti-GFP (#SC9996, Santa Cruz Biotechnology, Dallas, TX, USA), anti-LamB (#13435, Cell Signaling Technology, Danvers, MA, USA), and anti-β-actin (#A5441, Sigma-Aldrich, St. Louis, MO, USA).

### 4.4. Cell Culture, Transfection, and Reagents

CWR22Rv-1 and PC3 cell human prostate cancer cell lines were from ATCC, and the cells were grown in RPMI 1640 supplemented with 10% Fetal Bovine Serum (FBS, Sigma-Aldrich, St. Louis, MO, USA) and 1% penicillin/streptomycin at 37 °C with 5% CO_2_. Plasmid DNA was used to transfect cells using Plus/Lipofectamine reagent (Invitrogen, Carlsbad, CA, USA) for 24 h, according to the manufacturer’s instructions.

### 4.5. Western Blot Analysis

Briefly, protein concentration was determined using the Pierce BCA Protein Assay Reagent (Thermo Fisher Scientific, 23228, Waltham, MA, USA). From each sample, 30 μg protein was resolved by SDS-PAGE on Tris-glycine gels and transferred to polyvinylidene fluoride membranes. Membranes were blocked with 5% bovine serum albumin (B2064, Sigma-Aldrich, St. Louis, MO, USA) in Tris-buffered saline (T5030, Sigma-Aldrich, St. Louis, MO, USA) containing 0.1% Tween 20 (93773, Sigma-Aldrich, St. Louis, MO, USA) (TBST) and incubated with primary antibodies overnight at 4 °C. Membranes were washed three times for 5 min with TBST, incubated with either HRP-goat-anti-mouse (Abcam, ab6789, Cambridge, UK) or HRP-goat-anti-rabbit (Abcam, ab6721, Cambridge, UK) secondary antibodies for 1 h at room temperature. Blots were visualized using enhanced chemiluminescence (ECL from GE Healthcare, Braunschweig, Germany). Bands were quantified using Quantity One Software (Biorad, Hercules, CA, USA).

### 4.6. Immunoprecipitation

Cells were lysed with IGEPAL CA-630 buffer (50 mM Tris-HCl, pH 7.4, [T5030, Sigma-Aldrich, St. Louis, MO, USA], 1% IGEPAL CA-630 [I8896, Sigma-Aldrich, St. Louis, MO, USA], 10 mM EDTA, 150 mM NaCl, 50 mM NaF, 1 μM leupeptin [L5793, Sigma-Aldrich, St. Louis, MO, USA], and 0.1 μM aprotinin [SRE0050, Sigma-Aldrich, St. Louis, MO, USA]). Lysates were incubated with primary antibodies at 4 °C for 16 h and then incubated with protein G sepharose (GE healthcare, Uppsala, Sweden) at 4 °C for 3 h. Immunoprecipitated proteins were eluted at 95 °C for 2 min with 30 μL of 2× sample buffer (0.1 M Tris-HCl, pH6.8, 0.2 M DTT, 4% SDS, 20% glycerol, 0.2% bromophenol blue, and 1.43 M β-mercaptoethanol) and then used for immunoblotting.

### 4.7. CCK8 Assay

Transfected cells in suspension were digested using trypsin. Cell concentration was adjusted to 5000 cells/mL. Then 200 μL of this cell suspension was placed in 96-well plates. Each group had three parallel control wells. After the addition of CCK8 reagent (Thermo Fisher Scientific, Waltham, MA, USA), cells were incubated at 37 °C for 4 h. Plates were shaken for an additional 10 min. Absorbance values at 490 nm were then measured using a Microplate Reader (Bio-Rad Co., Singapore, Singapore).

### 4.8. Cell Fractionation

Nuclear and cytoplasmic proteins were fractionated using the Nuclear and Cytoplasmic Extraction Kit (Invitrogen, Thermo Fisher Scientific, Waltham, MA, USA) according to the manufacturer’s protocol. To separate the cytoplasmic and nuclear proteins, cell fractionation was performed. Briefly, cells were harvested with trypsin-EDTA and centrifuged at 500× *g* for 5 min. The supernatant was then removed carefully, and the cell pellet was washed with PBS followed by centrifugation at 500× *g* for 2–3 min. The supernatant was discarded carefully again to leave the cell pellet as dry as possible. Next, specific buffers provided by the Kit were added into tubes containing cell pellets to obtain cytoplasmic proteins and nuclear protein one by one. Protein extracts from different compartments of cells were subjected to immunoblotting. The lamin B antibody was used as a nuclear protein marker, while the β-actin antibody was used as a cytoplasmic protein marker.

### 4.9. Transwell Assay

A transwell assay was performed to evaluate cell invasion and migration abilities. Briefly, transfected Rv-1 cells were placed in the upper chamber with serum-free RPMI 1640, and the lower chamber was filled with culture medium supplemented with 0.5% fetal bovine serum (FBS) in the presence or absence of containing EGF (100 ng/mL). After incubating for 24 h, cells in the Matrigel (Sigma-Aldrich, St. Louis, MA, USA) were fixed with 4% paraformaldehyde and stained with 0.2% crystal violet (Beyotime, Shanghai, China). The cells were then observed under a light microscope and photographed. Ten fields were selected to count the number of cells to reflect cell mobility.

### 4.10. Cell migration Assay

Cell migration was measured using a scratch wound-healing assay. The same number of transfected cells were plated into six-well plates at a plating density sufficient to create a confluent monolayer after 12 h of culture at 37 °C in an incubator with 5% CO_2_. The monolayer was then scraped in a straight line with a P200 pipette tip to create a “scratch wound.” Images of wounded cell monolayers were taken under a Model IX70 Microscope (Olympus, Tokyo, Japan) at 0 h and 72 h after wounding. Cell migration into the wounded area was recorded using the same microscope equipped with a CoolSNAP HQ CCD Camera (Nippon Roper, Chiba, Japan) and MetaMorph Software (Universal Imaging Co., Ltd., Buckinghamshire, UK). The healing rate was quantified using measurements of gap size after culture. Five different areas in each assay were chosen to measure the distance of migrating cells to the origin of the wound edge. The distance and the wound edge were measured using the “measurement length” function in Image J Software (National Institutes of Health, Bethesda, MD, USA).

### 4.11. qRT-PCR

RNA isolation, a semi-quantitative Reverse Transcriptase PCR and a quantitative Real-Time Reverse Transcription PCR were performed. Total RNAs were isolated from cells using TRIzol Reagent (15596018; Invitrogen, Carlsbad, CA, USA). cDNA was then synthesized using a high-capacity cDNA synthesis kit (4374967; Thermo Fisher Scientific, Waltham, MA, USA). Subsequently, semi-quantitative Reverse Transcriptase PCR (sqRT-PCR) was conducted in a thermal cycler (T100, Bio-Rad, Hercules, CA, USA). PCR signal intensity of each gene was electrophoresed in agarose gels and visualized with EtBr-Imaging (GelDoc XR+; Bio-Rad, Hercules, CA, USA). In addition, a quantitative Real-Time PCR (qRT-PCR) amplification was performed for indicated genes using SYBR Green (04887352001, Roche, Basel, Switzerland). PCR thermal cycling involved a denaturing step at 95 °C for 10 min, followed by 45 cycles of annealing at 95 °C for 10 s, 60 °C for 10 s and 72 °C for 20 s. The target gene expression was normalized to GAPDH gene expression.

p21 Forward Primer: CGATGGAACTTCGACTTTGTCA;

Reverse Primer: GCACAAGGGTACAAGACAGTG.

GAPDH Forward Primer: GGAGCGAGATCCCTCCAAAAT; Reverse Primer: GGCTGTTGTCATACTTCTCATGG.)

### 4.12. Immunofluorescence

The cells were cultured on sterile glass coverslips in six-well plates and fixed with PBS containing 3.7% paraformaldehyde for 15 min at room temperature. Cells were permeabilized with PBS containing 0.2% Triton X-100. Indicated antibodies were incubated overnight at 4 °C. Secondary antibodies were coupled to Alexa 647 (Abcam). Nuclear counterstaining was performed with DAPI (Invitrogen). Additionally, all images were acquired using Leica TCS SP5 Confocal microscope (Leica, Wetzlar, Germany).

### 4.13. Statistical Analysis

All statistical analyses were performed with GraphPad Prism 8 (GraphPad, San Diego, CA, USA). All data are expressed as mean ± SE. They were evaluated by ANOVA with a student’s *t*-test. Comparison between two groups was performed by a student’s *t*-test. One-way or two-way analysis of variance (ANOVA) with post-Bonferroni corrections was used to compare the effect of different treatments or compare the effect of treatment along with time. A value of * *p* < 0.01 denotes statistical significance. As for the Single Exponential Decay Model, blots data were quantified using Quantity One Software (Biorad). A linear model was used for statistics. One phase exponential decay modeling of protein degradation rate can be determined with the following equation: N(t) = N0e^−λt^, where λ was the exponential decay constant, and span was the difference between the initial data of protein and the plateau of protein level.

## Figures and Tables

**Figure 1 ijms-22-05306-f001:**
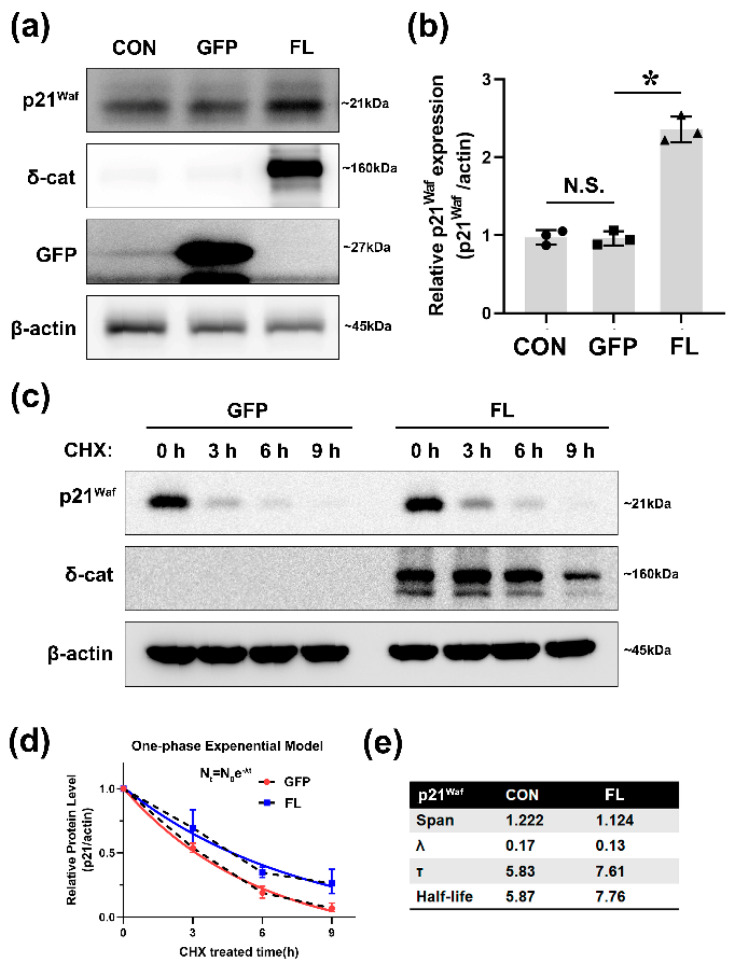
δ-catenin increases p21 protein level by promoting its stabilization. (**a**) Rv-1 cells were transfected with GFP, full-length δ-catenin-GFP (FL), or un-transfected. Cell lysates were immunoblotted with anti-p21^waf^, anti-δ-catenin, anti-GFP, and anti-β-actin. This experiment was repeated independently three times. (**b**) The quantitation of p21^waf^ shown in (**a**,**c**) Rv-1 cells were transfected with the respective plasmid followed with treating with 100 μg/mL of cycloheximide (CHX). Cells were harvested at different time points and cell lysates were immunoblotted with antibodies to p21^waf^, δ-catenin, and actin. (**d**,**e**) The quantification of p21 protein levels in (**c**). One phase exponential decay model was used for analyzing the stabilization of p21 with the following equation: N(t) = N0e^−λt^, where λ was the exponential decay constant. Span was the difference between the initial protein level of p21 and the plateau of p21. The fit of curves was significantly different (*Mann–Whitney test, p <* 0.0001). Dotted line: Linear model. All *p*-values were determined with unpaired Student’s *t*-test: * *p* < 0.001.

**Figure 2 ijms-22-05306-f002:**
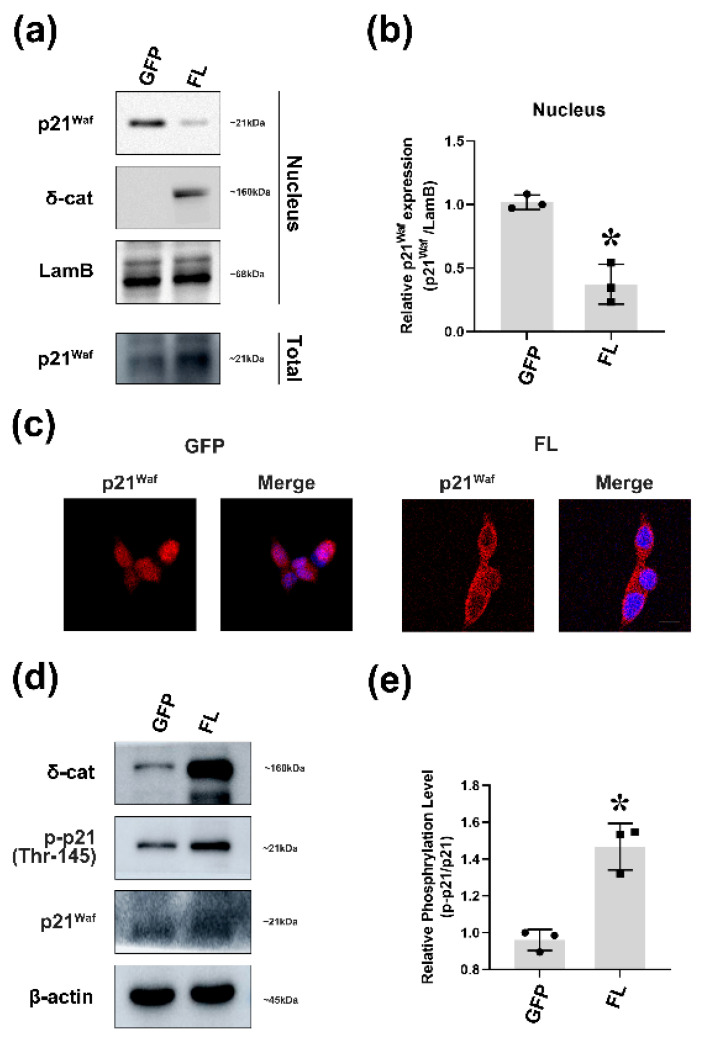
δ-catenin blocks p21 nuclear accumulation through stabilization of p21 in the cytoplasm. (**a**) Rv-1 cells were transfected with GFP or GFP-δ-catenin. Nuclear p21 levels were measured by immunoblotting with anti-p21^waf^, anti-δ-catenin, anti-LamB. This experiment was repeated independently three times. (**b**) Densitometry data of p21 from (**a**) were normalized using LamB (Nucleus) densitometry data. Quantitate p21/LamB. * *p* < 0.001 vs. GFP (**c**) Immunofluorescence images of Rv-1 cells transfected with respective plasmid were labeled with indicated antibodies against p21. Sections were co-stained with DAPI to visualize nuclei. Scar bar: 100 μm. (**d**) Rv-1 cells were transfected with GFP or δ-catenin-GFP. Cell lysates were immunoblotted with anti-p-p21 (Thr-145), anti-p21^waf^, anti-δ-catenin, and anti-β-actin. This experiment was repeated independently three times. (**e**) Quantitation of p-p21/p21 in (**d**). * *p* < 0.001 vs. GFP. All *p*-values were determined with unpaired Student’s *t*-test: *, *p* < 0.001.

**Figure 3 ijms-22-05306-f003:**
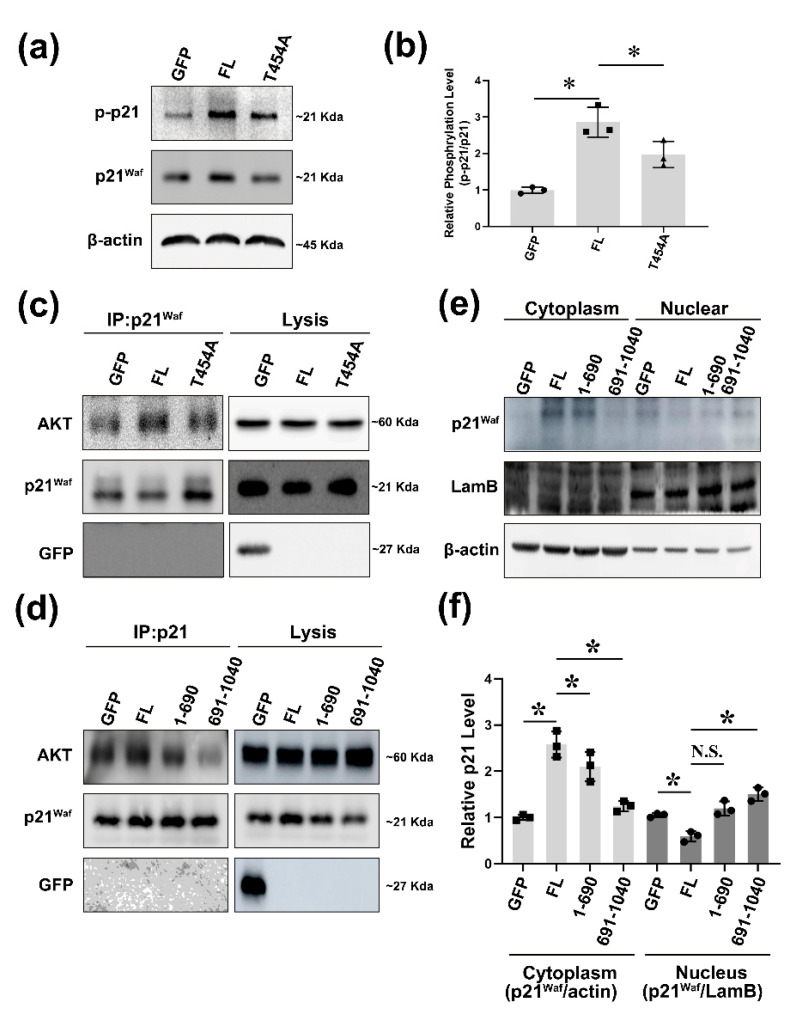
δ-catenin promotes the interaction between p21 and AKT, which phosphorylates p21 at Thr-145 residue. (**a**) Rv-1 cells were transfected with GFP, δ-catenin-GFP, or GFP-T454A. Cell lysates were immunoblotted with anti-p21^waf^, anti-p-p21, and anti-β-actin. This experiment was repeated independently three times. (**b**) The quantitation of p-p21/p21 in (**a**,**c**) Immunoprecipitation with anti-p21^waf^ and immunoblotting with anti-AKT of lysates from Rv-1 cells transfected with the respective plasmid. (**d**) Immunoprecipitation with anti-p21^waf^ and immunoblotting with anti-AKT of lysates from Rv-1 cells transfected with the respective plasmid. (**e**) Rv-1 cells were transfected with respective plasmid. Nuclear and cytoplasmic p21 levels were measured by immunoblotting with anti-p21^waf^, anti-LaminB1, and anti-β-actin. This experiment was repeated independently three times. (**f**) Densitometry data of p21 in (**e**) were normalized using LamB (Nucleus) or β-actin (Cytoplasm) densitometry data. All *p*-values were determined with unpaired Student’s *t*-test: *, *p* < 0.001.

**Figure 4 ijms-22-05306-f004:**
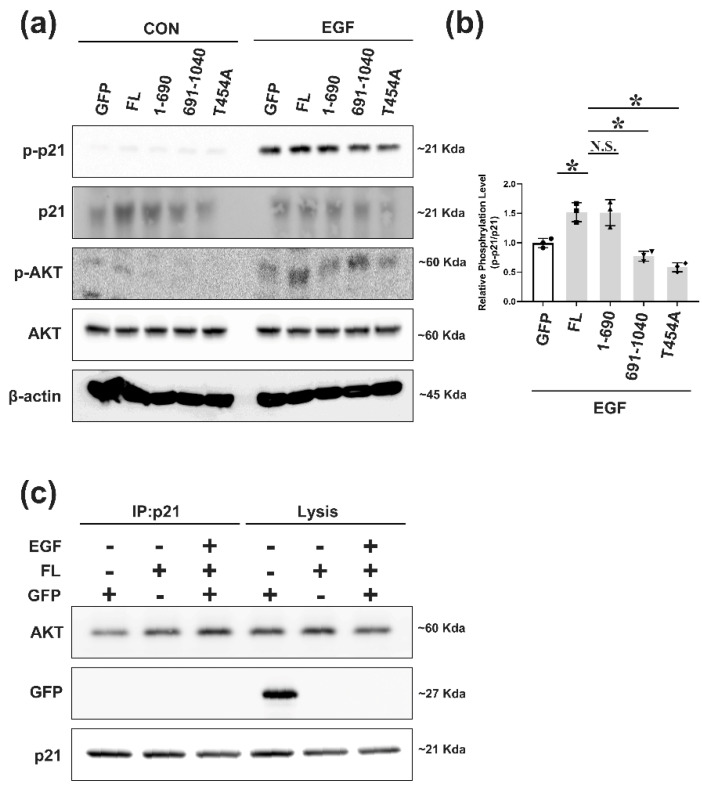
EGF is one upstream ligand for regulating AKT/δ-catenin/p21 signaling. (**a**) Rv-1 cells were transfected with the respective plasmid followed by serum starvation and treated with EGF 100 ng/mL for 5 min. This experiment was repeated independently three times. (**b**) The quantitation of p-p21/p21 in (**a**,**c**). Immunoprecipitation with the anti-p21^waf^ antibody, followed by immunoblotting with the anti-AKT antibody using lysate obtained from transfected Rv-1 cells after serum starvation treatment with EGF 100 ng/mL for 5 min. All *p*-values were determined with unpaired Student’s *t*-test: *, *p* < 0.001.

**Figure 5 ijms-22-05306-f005:**
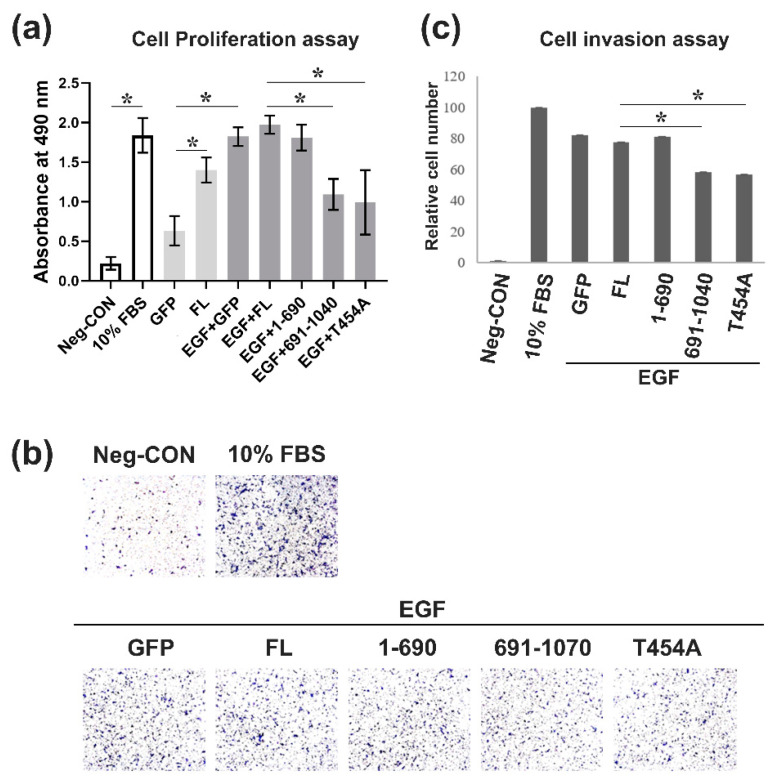
EGF regulates cell proliferation and invasion through AKT/δ-catenin/p21 signaling independently. (**a**) Rv-1 cells were transfected with the respective plasmid and then seeded into a 96-well plate. After 12 h, transfected cells were then serum-starved and then treated with 100 ng/mL EGF or 10% serum for 12 h. After cells were incubated with the CCK8 reagent at 37 °C for 4 h, plates were read at 490 nm to obtain absorbance values. This experiment was repeated independently five times. (**b**) A transwell assay was performed with Rv-1 cells transfected with the respective plasmid and treated with EGF 100 ng/mL. This experiment was repeated independently five times. (**c**) The quantitation results of positive cells are shown in (**b**). All *p*-values were determined with unpaired Student’s *t*-test: *, *p* < 0.001.

**Figure 6 ijms-22-05306-f006:**
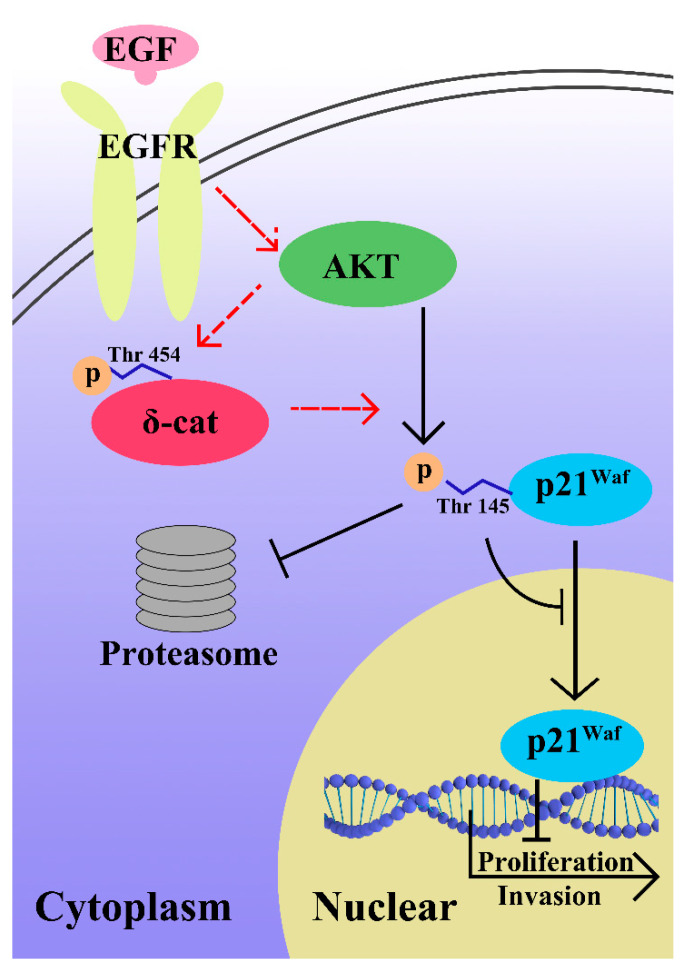
Schematic model showing the role of δ-catenin in EGF/AKT/p21 signaling in prostate cancer.

## Data Availability

The data presented in this study are available on request from the corresponding author.

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
