# Peer review of "δ-Catenin Participates in EGF/AKT/p21Waf Signaling and Induces Prostate Cancer Cell Proliferation and Invasion"

_ijms, 2021, doi:10.3390/ijms22105306_

Round 1
Reviewer 1 Report
In the present study, is indicated a novel role of δ-catenin in prostate cancer progression through its participation in the EGF/Akt/p21 axis. Specifically, authors provide evidence that δ-catenin is capable of stabilizing p21 cytoplasmic localization and regulating Akt and p21 interaction, while a role of EGF as an upstream regulator of Akt/δ-catenin/p21, in promoting proliferative and migratory/invasive potential of human prostate cancer cells CWR22Rv-1, is described. Results are interesting, further supporting previous data by the same group, relevant to different underlying molecular mechanisms of δ-catenin-mediated prostate cancer progression, such as alteration of cell cycle and survival gene profiles, induction of E-cadherin processing and enhancement of EGFR/ERK1/2 signaling pathway.
Below are some parts of the manuscript that need to be addressed
Introduction
Introduction is generally well written, describing the distinct roles of δ-catenin, p21 and ultimately EGF/EGFR in prostate cancer progression. I recommend that authors should rephrase lines 24-25, while chronic inflammation could be added as an additional predisposing factor to prostate cancer onset (Lines 28-30). In addition:
Lines 52-53 should be rephrased
Line 58: Please explain the biological significance of p21 nuclear accumulation blockage, as opposed to that of p21 cytoplasmic stabilization
Line 60: Please correct epidermis growth factor to: epidermal growth factor
Line 75: Please correct cytoplasm p21 to: cytoplasmic p21
Results
Line 96: Please replace Figure 1b with 1c, as this is the correct figure according to relevant text. In addition, authors should describe to a greater extent the single exponential decay model (preferentially in a distinct section in Materials and Methods), in order to be clear for the readers
Lines 107, 145 175, 205-206: It is not evident that this is a quantification of membrane bound p21 protein. Please explain/rephrase
Lines 134-135: Supplementary Figures 3a/3b indicate total nuclear and not phosphorylated p21 levels, while Sup Figures 3c/3d do not indicate nuclear accumulation of p21. Please correct
Line 152: LY inhibitor is not an Akt inhibitor, but a PI3K inhibitor. Please correct. Please use the full name of LY inhibitor (LY294002) throughout the manuscript
Lines 152-153: In Sup Figure 2 are observed phosphorylated levels of p21, while cytoplasmic p21 appears unaffected following LY incubation
Figure 3a and 3b: Although in Fig.3a it evident that p-p21 levels in T454A are higher compared to those in GFP sample, in Fig.3b quantification appears significantly reduced. How is this explained by the authors?
Figure 3d: Blots for p21 are over-exposed and should be replaced
Figure 4a. It is evident that also in the case of 691-1040 mutant, pAkt levels appear to be induced. How is this result explained by the authors?
Lines 232-236: Please rephrase
Discussion
Addition of a more detailed paragraph stating the biological significance of the results, in the specific context of therapeutic targeting of EGF(R)/Akt/p21/δ-catenin axis in prostate cancer, should be really helpful
Materials and Methods
Line 308: LY294002 is a PI3K inhibitor. Please correct
4.8 Cell fractionation: Please provide information relevant to ‘’Kit’’ used for the differential fractionation procedure (cytoplasmic, nuclear and membrane extracts). In addition although it is stated that E-cadherin was used as a membrane protein marker, relevant blots are missing throughout the entire manuscript (as membrane bound p-21 markers)
Finally, a large amount of unpublished data are provided but not mentioned/described by the authors, throughout the entire manuscript
Author Response
Introduction is generally well written, describing the distinct roles of δ-catenin, p21 and ultimately EGF/EGFR in prostate cancer progression. I recommend that authors should rephrase lines 24-25, while chronic inflammation could be added as an additional predisposing factor to prostate cancer onset (Lines 28-30). In addition:
[Answer] Thank you for your advice. Considering that the meaning of this sentence (line 24-25) coincides with the previous one (line 23-24), we deleted this sentence to avoid duplication. As for lines 28-30, “African descent” was replaced by "chronic inflammation”
Lines 52-53 should be rephrased
[Answer] This phrase was modified according to the comment (line 50-55). “It plays key roles in various cellular evets like apoptosis, cell migration, DNA repair, suppressing tumor formation. In response to DNA damage or other cellular stressors, p21 expression can be increased, resulting in the activation of cell cycle checkpoints until repair takes place.”
Line 58: Please explain the biological significance of p21 nuclear accumulation blockage, as opposed to that of p21 cytoplasmic stabilization
[Answer] We have modified the sentence according to the comment (line 55-61). “However, resent research studies have revealed that p21 not only functions as a tumor suppressor, but also acts as an oncogene with a dual behavior in different cellular location. In many cancer types, p21 is stabilized in cytoplasm to inhibit its nucleus translocation. Moreover, accumulation of p21 in the cytoplasm may induce cell proliferation and suppresses apoptosis, and further lead to resistance against chemotherapy and radiotherapy.”
Line 60: Please correct epidermis growth factor to: epidermal growth factor
[Answer] Thank you for underlining this deficiency. Epidermis was revised.
Line 75: Please correct cytoplasm p21 to: cytoplasmic p21
[Answer] We apology for the language errors. This section was modified.
Results
Line 96: Please replace Figure 1b with 1c, as this is the correct figure according to relevant text. In addition, authors should describe to a greater extent the single exponential decay model (preferentially in a distinct section in Materials and Methods), in order to be clear for the readers
[Answer] Thank you for finding this deficiency. Figure 1b. Figure 1c, Figure 1d and Figure 1e were now in the right place. As for the reviewer’s concern, we have updated single exponential decay model details in the Materials and Methods.
Lines 107, 145 175, 205-206: It is not evident that this is a quantification of membrane bound p21 protein. Please explain/rephrase
[Answer] We thank the reviewed for the advice. In fact, “Quantification of membrane bound p21 protein” was removed.
Lines 134-135: Supplementary Figures 3a/3b indicate total nuclear and not phosphorylated p21 levels, while Sup Figures 3c/3d do not indicate nuclear accumulation of p21. Please correct
[Answer] We apologize for the incorrect description. Lines 134-135 was modified to “Results revealed that silencing δ-catenin in Rv-1 cells also showed a strong promoted effect on nuclear accumulation (supplement Figure 3a and 3b) and block the phosphorylation of p21 (supplement Figures 3c and 3d).”
Line 152: LY inhibitor is not an Akt inhibitor, but a PI3K inhibitor. Please correct. Please use the full name of LY inhibitor (LY294002) throughout the manuscript
[Answer] Thank you for your advice. In fact, because PI3K is upstream of Akt, LY294002 is also often used to inhibit the activity of Akt, that is the reason why described LY294002 as an inhibitor of Akt. It has now been corrected.
Lines 152-153: In Sup Figure 2 are observed phosphorylated levels of p21, while cytoplasmic p21 appears unaffected following LY incubation
[Answer] We have modified the sentence according to the comment (line 160-161).
Figure 3a and 3b: Although in Fig.3a it evident that p-p21 levels in T454A are higher compared to those in GFP sample, in Fig.3b quantification appears significantly reduced. How is this explained by the authors?
[Answer] We are grateful for the suggestion. There was a mistake in our statistical process. Now we have re-counted 3a and corrected the Figure 3b.
Figure 3d: Blots for p21 are over-exposed and should be replace.
[Answer] Thank you for the review’s concern, we have re-exposed the blots for p21.
Figure 4a. It is evident that also in the case of 691-1040 mutant, pAkt levels appear to be induced. How is this result explained by the authors?
[Answer] We thank the reviewer for the very interesting comment. In fact, we are also curious about this phenomenon. However, the current researches cannot reveal the regulation mode of δ-catenin over AKT. As for establishing this relationship we need to purchase some related reagents and carry out a lot of experiments for exploration and verification. Considering the adjustment of AKT to δ-catenin, we guess this may be related to feedback regulation in the organism. Therefore, it is particularly important for us to explore the relationship between δ-catenin and AKT in future studies. Therefore, we seek for the reviewer’s understanding.
Lines 232-236: Please rephrase
[Answer] We have modified the sentence according to the previous comment. And modified sentences were “Together, our finding provide compelling evidence that the pro-proliferation and pro-migration effect of EGF depends on the presence of 1-690 or T454 phosphorylation of δ-catenin which closely regulated by AKT. Overall, these results are not difficult to find considering the previous results that AKT/δ-catenin/p21 signaling plays an important role in the regulation of EGF on cell proliferation and invasion.” in line 238-245.
Discussion
Addition of a more detailed paragraph stating the biological significance of the results, in the specific context of therapeutic targeting of EGF(R)/Akt/p21/δ-catenin axis in prostate cancer, should be really helpful
[Answer] Thank you for your suggestion. We have added the information about the therapeutic significance of targeting EGF(R)/Akt/p21/δ-catenin axis in prostate cancer in line 312-316.
Materials and Methods
Line 308: LY294002 is a PI3K inhibitor. Please correct.
[Answer] This section was modified.
4.8 Cell fractionation: Please provide information relevant to ‘’Kit’’ used for the differential fractionation procedure (cytoplasmic, nuclear and membrane extracts). In addition although it is stated that E-cadherin was used as a membrane protein marker, relevant blots are missing throughout the entire manuscript (as membrane bound p-21 markers)
[Answer] We are very sorry for the mistakes in this manuscript and inconvenience they caused. We have revised the materials and methods, corrected and improved the relevant errors you mentioned.
Reviewer 2 Report
This research Manuscript found that δ-catenin could enhance the p21waf phosphorylation, which results in the stabilization of p21waf in the cytoplasm and decrease of nuclear accumulation. This research also investigates the relation of these proteins within one prostate cancer cell line Rv-1: EGF, AKT, δ-catenin, and p21waf.
I have the following questions/concerns which need to be addressed by the authors. It will be necessary that the authors conduct some experiments to address these questions/concerns.
Main Points:
- In some western blot of Rv-1 control cells or Rv-1 transfected with GFP (Fig. 2C, S3C), the δ-catenin can be detected, whereas, in some western blot of Rv-1 control cells or Rv-1 transfected with GFP (Fig. 1a ), the δ-catenin seems almost cannot be detected. Please address this concern.
- In Figure 2a, 2b, it’s important that the cytoplasmic level of p21waf was also shown by western blot and quantified. Besides, it’s also important to also shown the cytoplasmic level δ-catenin with western blot. In Fig 2a, nucleus level δ-catenin is almost not existing in GFP but exists in GFP-δ. It’s important to know the cytoplasmic δ-catenin level of these cells.
- Have authors measured the effect of δ-catenin T454A on the p21waf level in the nucleus (related to Fig. 3). This result is important because it provides information related to the importance of this residue (T454) on the function of δ-catenin on p21waf . This result related to mutant δ-catenin T454A also could reconfirm the effect of δ-catenin on p21waf .
- In Figure 2, it’s better if authors could use another experiment design to confirm this result, for instance, use immunofluorescence assay to show the nuclear level of p21waf.
- In Fig 2 and 3, since authors generate these δ-catenin mutants, it’s better to also include a western blot to display all mutants δ-catenin to confirm the expression of mutants δ-catenin (at least in supplemental Figure).
- It’s better if authors can add a Schematic illustration that shows the deletion region or mutant site of all δ-Catenin variants generated in this study.
- The RTK receptor of EGF is EGFR. In previous authors also showed that δ-Catenin Increases the Stability of EGFR and Enhances EGFR/Erk1/2 Signaling in Prostate Cancer, I think that authors should conduct these studies related to EGFR to answer these questions.
1) Have authors measured the Effect of δ-Catenin on the Stability of EGFR in this research?
2)Have authors measured the membrane-bound EGFR protein level?
Because activated EGFR can also activate AKT pathways (beside ERK pathway) if δ-Catenin could also stabilize EGFR. The effect of δ-Catenin on p21waf may also from this effect (at least partly). It is necessary to address this concern.
- In line 293, the authors claim that “EGF could phosphorylate AKT”. Please accurately wrote this sentence. Correct me if I am wrong, EGF does not directly phosphorylate AKT, EGF activates EGFR signaling pathways which could phosphorylate AKT. Have authors measured the effect of EGF on the phosphorylation and activity of EGFR? It’s necessary to check and show this information.
- In Fig. 3 b and 3a, the quantification result of P-p21/p21 in T454A seems are rapidly less than GFP control in Fig. 3b, but not in western blot Fig. 3a. These two panels are not consistent. Please address this concern.
- In addition to studying the effect of δ-catenin on p21 with single cell lines (prostate cancer cell line Rv-1), it’s important to include at least another prostate cancer cell line in this research paper to confirm that this discovery could be found in multiple prostate cancer cell line.
Authors can conduct the molecular biology investigation and cellular study (proliferation/invasion assay) after the expression of δ-catenin in the second prostate cancer cell line. Authors can also check the effect of δ-catenin knock-down on p21waf, P-p21waf (Thr-145) and p21waf level in the nucleus using a cancer cell line originally express relative high level of δ-catenin.
Minor points
- Authors must add more detailed information in the section of material and method because most of the methods are not detailed enough. Just list a few examples. In “4.6. Immunoprecipitation”, what is the composition of lysate buffer used in this experiment? In “4.9. Transwell assay”, how long is the incubation time for this cell invasion assay? This information is important.
- The antibody of pAKT in this study is related to Phospho-Akt (Ser 473), not other sites (Thr 308), please clearly wrote the phosphorylation site of AKT (Ser 473) in the manuscript.
Author Response
Main Points:
- In some western blot of Rv-1 control cells or Rv-1 transfected with GFP (Fig. 2C, S3C), the δ-catenin can be detected, whereas, in some western blot of Rv-1 control cells or Rv-1 transfected with GFP (Fig. 1a), the δ-catenin seems almost cannot be detected. Please address this concern.
[Answer] Thank you for your question. As for the reviewer’s concern, in Figure 1a, the δ-catenin seems almost cannot be detected, we think this may be related to our different exposure time. Due to the selection of automatic exposure mode, the exposure time of our different membranes may not be consistent, and we will pay attention to this problem in the subsequent experiments.
- In Figure 2a, 2b, it’s important that the cytoplasmic level of p21waf was also shown by western blot and quantified. Besides, it’s also important to also shown the cytoplasmic level δ-catenin with western blot. In Fig 2a, nucleus level δ-catenin is almost not existing in GFP but exists in GFP-δ. It’s important to know the cytoplasmic δ-catenin level of these cells.
[Answer] Thank you for pointing out this question. In fact, the reason why we did not detect the protein content of p21 in the cytoplasm is considering the regulation of δ-catenin for p21 stabilization. In addition, p21 in total lysis was also used to support the role of δ-catenin in nuclear accumulation of p21. As for δ-catenin in cytoplasm, we have detected the distribution of δ-catenin in prostate cancer cell in previous studies[1], and found that under normal circumstances, it is mainly concentrated in the cytoplasm. The reason why there is relatively large amounts of δ-catenin in the nucleus can be due to the fact that we transfected the plasmid of δ-catenin, and it did have a specific nuclear localization sequence
- Have authors measured the effect of δ-catenin T454A on the p21waf level in the nucleus (related to Fig. 3). This result is important because it provides information related to the importance of this residue (T454) on the function of δ-catenin on p21waf. This result related to mutant δ-catenin T454A also could reconfirm the effect of δ-catenin on p21waf.
[Answer] Thank you for your suggestion. Considering the Reviewer’s suggestion, we have re-done the effect of δ-catenin T454A on the p21waf level in the nucleus. And we confirmed that δ-catenin T454A can induce p21waf nucleus accumulation (Supplement Figure S3e)
- In Figure 2, it’s better if authors could use another experiment design to confirm this result, for instance, use immunofluorescence assay to show the nuclear level of p21waf.
[Answer] We thank the reviewer for the comment. In fact, we do the immunofluorescence assay to show the nuclear level of p21waf in different δ-catenin mutated cells (Figure 2c)
- In Fig 2 and 3, since authors generate these δ-catenin mutants, it’s better to also include a western blot to display all mutants δ-catenin to confirm the expression of mutant δ-catenin (at least in supplemental Figure).
- It’s better if authors can add a Schematic illustration that shows the deletion region or mutant site of all δ-Catenin variants generated in this study.
[Answer] We are very sorry for our negligence. Actually, we have displayed all mutant δ-catenin by western blot. And we showed the schematic illustration and western blot in the supplementary Figure 1.
- The RTK receptor of EGF is EGFR. In previous authors also showed that δ-Catenin Increases the Stability of EGFR and Enhances EGFR/Erk1/2 Signaling in Prostate Cancer, I think that authors should conduct these studies related to EGFR to answer these questions.
1) Have authors measured the Effect of δ-Catenin on the Stability of EGFR in this research?
2) Have authors measured the membrane-bound EGFR protein level?
Because activated EGFR can also activate AKT pathways (beside ERK pathway) if δ-Catenin could also stabilize EGFR. The effect of δ-Catenin on p21waf may also from this effect (at least partly). It is necessary to address this concern.
[Answer] Thank you very much for your professional advice. The studies that are mentioned in Review’s comment was similar to a research what we’ve published before[2]. In fact, we have considered before the initial submission of this manuscript. These are, without a doubt, δ-catenin/EGFR interactions and the reciprocal regulation would not rule out a role for additional function of δ-catenin to AKT/p21. Moreover, from δ-catenin T454A mutant data provide evidence for the regulation of δ-catenin in AKT/p21.
In line 293, the authors claim that “EGF could phosphorylate AKT”. Please accurately wrote this sentence. Correct me if I am wrong, EGF does not directly phosphorylate AKT, EGF activates EGFR signaling pathways which could phosphorylate AKT. Have authors measured the effect of EGF on the phosphorylation and activity of EGFR? It’s necessary to check and show this information.
[Answer] We have made correction according to the Reviewer’s comments. As Reviewer suggested that showing relationship between EGF/EGFR/AKT, we treated Rv-1 with EGFR inhibitor (Gefitinib) to analyze the function of EGF/EGFR axis in p21 phosphorylation.
- In Fig. 3 b and 3a, the quantification result of P-p21/p21 in T454A seems are rapidly less than GFP control in Fig. 3b, but not in western blot Fig. 3a. These two panels are not consistent. Please address this concern.
[Answer] We are grateful for the suggestion. There was a mistake in our statistical process. Now we have re-counted 3a and corrected the Figure 3b.
- In addition to studying the effect of δ-catenin on p21 with single cell lines (prostate cancer cell line Rv-1), it’s important to include at least another prostate cancer cell line in this research paper to confirm that this discovery could be found in multiple prostate cancer cell line.
Authors can conduct the molecular biology investigation and cellular study (proliferation/invasion assay) after the expression of δ-catenin in the second prostate cancer cell line. Authors can also check the effect of δ-catenin knock-down on p21waf, P-p21waf (Thr-145) and p21waf level in the nucleus using a cancer cell line originally express relative high level of δ-catenin.
[Answer] We are grate for the suggestion. To be more clear with the reviewer concerns, we use another PC-derived cell line(PC3), and treated with EGF, si-δ-catenin, and δ-cateninT454A to confirmed the function of δ-catenin in p21 nucleus accumulation(Supplement 6).
Minor points
- Authors must add more detailed information in the section of material and method because most of the methods are not detailed enough. Just list a few examples. In “4.6. Immunoprecipitation”, what is the composition of lysate buffer used in this experiment? In “4.9. Transwell assay”, how long is the incubation time for this cell invasion assay? This information is important.
[Answer] We are very sorry for our negligence of the section of material and methods. As for the reviewer’s concern, the full descriptions of material and methods like “Immunoprecipitation”, “Transwell assay”, and so on.
- The antibody of pAKT in this study is related to Phospho-Akt (Ser 473), not other sites (Thr 308), please clearly wrote the phosphorylation site of AKT (Ser 473) in the manuscript.
[Answer] As for the reviewer’s concern, we have corrected the antibody of pAKT to Phospho-Akt (Ser 473).
- Li M, Nopparat J, Aguilar BJ, Chen YH, Zhang J, Du J, Ai X, Luo Y, Jiang Y, Boykin C et al: Intratumor delta-catenin heterogeneity driven by genomic rearrangement dictates growth factor dependent prostate cancer progression. Oncogene 2020, 39(22):4358-4374.
- Shrestha N, Shrestha H, Ryu T, Kim H, Simkhada S, Cho YC, Park SY, Cho S, Lee KY, Lee JH et al: delta-Catenin Increases the Stability of EGFR by Decreasing c-Cbl Interaction and Enhances EGFR/Erk1/2 Signaling in Prostate Cancer. Mol Cells 2018, 41(4):320-330.
Round 2
Reviewer 1 Report
The revised version of the manuscript has been significantly improved. Authors have provided a number of details relevant to the previous recommendations. My only suggestions are to include information relevant to PC3 cell line (media used) in ‘’Cell culture, transfection and reagents’’ section, as well as to indicate the meaning (at least in the first appearance in the manuscript of the abbreviation FL observed in different figures). Moreover, authors are encouraged to modify the sequence of all S6 figures, placing blots and relevant quantitation graphs in the same line (as well as for the transwell assay). Finally, in S6 figure legend, please correct in last line: (g) Quantitation of positive cells shown in (b) to: shown in (f).
Author Response
[Answer]: We are sorry for our oversight. This has been corrected it in the revised manuscript. In Materials and Methods, we now add PC3 cell culture methods. Moreover, we have spelled out the abbreviation FL (full length) in the first appearance. According to the comments of the reviewer, we have also made an adjustment the figure S6. Finally, we have corrected the S6 figure legend. We attached the file for the certificate of professional English editing.

Reviewer 2 Report
The authors adequately address my concerns and questions. I don't have more questions. Please double-check the English language and any typos.
Author Response
[Answer]: We are very sorry for the problem in the previous version, if this caused you difficulty in understanding. We sent the previous version to a professional editor since English is not our native language and improve the language for the manuscript. We attached the file for the certificate of English editing.

This manuscript is a resubmission of an earlier submission. The following is a list of the peer review reports and author responses from that submission.